# Implementation of infection prevention and control practices in an upcoming COVID-19 hospital in India: An opportunity not missed

**Arghya Das[1], Rahul Garg[1], E. Sampath Kumar[2], Dharanidhar Singh[2], Bisweswar Ojha[3], H. Larikyrpang Kharchandy[1], Bhairav Kumar Pathak[3], Pushkar Srikrishnan[2], Ravindra Singh[4], Immanuel Joshua[2], Sanket Nandekar[2], Vinothini J.[2], Reenu Reghu[2], Nikitha Pedapanga[2], Tuhina Banerjee[1]\*, Kamal Kumar Yadav[1]**

**1** Department of Microbiology, Institute of Medical Sciences, BHU, Varanasi, India, **2** Department of Community Medicine, Institute of Medical Sciences, BHU, Varanasi, India, **3** Department of Pharmacology, Institute of Medical Sciences, BHU, Varanasi, India, **4** Trauma Centre, Institute of Medical Sciences, BHU, Varanasi, India

\* drtuhina@yahoo.com

**Data Availability Statement:** All relevant data are already present in the manuscript and additional supporting data file of this qualitative research

## Abstract

Infection prevention and control (IPC) program is obligatory for delivering quality services in any healthcare setup. Lack of administrative support and resource-constraints (under-staffing, inadequate funds) were primary barriers to successful implementation of IPC practices in majority of the hospitals in the developing countries. The Coronavirus Disease 2019 (COVID-19) brought a unique opportunity to improve the IPC program in these hospitals. A PDSA (Plan—Do—Study- Act) model was adopted for this study in a tertiary care hospital which was converted into a dedicated COVID-19 treatment facility in Varanasi, India. The initial focus was to identify the deficiencies in existing IPC practices and perceive the opportunities for improvement. Repeated IPC training (induction and reinforce) was conducted for the healthcare personnel (HCP) and practices were monitored by direct observation and closed-circuit television. Cleaning audits were performed by visual inspection, review of the checklists and qualitative assessment of the viewpoints of the HCP was carried out by the feedbacks received at the end of the training sessions. A total of 2552 HCP and 548 medical students were trained in IPC through multiple offline/onsite sessions over a period of 15 months during the ongoing pandemic. Although the overall compliance to surface disinfection and cleaning increased from 50% to >80% with repeated training, compliance decreased whenever newly recruited HCP were posted. Fear psychosis in the pandemic was the greatest facilitator for adopting the IPC practices. Continuous wearing of personal protective equipment for long duration, dissatisfaction with the duty rosters as well as continuous posting in high-risk areas were the major obstacles to the implementation of IPC norms. Recognising the role of an infection control team, repeated training, monitoring and improvisation of the existing resources are keys for successful implementation of IPC practices in hospitals during the COVID-19 pandemic.

could not be uploaded due to ethical restrictions for free access. However, all data may be made available on official request from authentic sources. The request should be channeled through the Infection Prevention and Control Committee, COVID-19 Task Force, Institute of Medical Sciences, Banaras Hindu University, Varanasi-221005 (Contact Email ID: infectioncontrolteamimsbhu@gmail.com).

**Funding:** The author(s) received no specific funding for this work.

**Competing interests:** The authors have declared that no competing interests exist.

## Introduction

Developing countries are often challenged with the implementation of infection prevention and control (IPC) practices in their healthcare systems. The World Health Organization (WHO) also recommends the implementation of IPC programs in every acute healthcare facility owing to the substantial evidence on the decrease in healthcare-associated infections in association with the effective functioning of the IPC programs [1]. However, despite these recommendations, the majority of the tertiary care hospitals are far from implementing IPC programs in their setup. Other than a few accredited healthcare organizations, organized IPC programs are not routinely practiced in India. Recently, the global pandemic of Coronavirus Disease 2019 (COVID-19) had put forward several challenges. Besides the highly infectious nature of the disease and mortality (according to the WHO estimate, a total of 3,059,642 confirmed cases of infection and 211,028 deaths were reported from various parts of the world in the first four months of the pandemic itself) [2], lack of appropriate preventive measures in form of sufficient vaccine therapy or treatment in severely ill patients in the initial phase of the pandemic has prioritized the rapid implementation of IPC practices as the last resort for protection against the highly transmissible severe acute respiratory syndrome coronavirus 2 (SARS-CoV-2). In this context, we describe the setting up and execution of IPC practices in an existing tertiary care hospital in north India during its transformation to a COVID-19 hospital and assess the different factors influencing the promotion and implementation of the IPC practices.

## Materials and methods

### Healthcare facility

The proposed healthcare setup, where the IPC practices were to be implemented, was an existing more than 1500 bedded tertiary care hospital in Varanasi, North India. The catchment area of the hospital was considerably large as it is the premier tertiary care hospital providing specialty services to the health care needs of about 2 billion population of more than 5 states in India as well as the neighbouring country of Nepal.

### Ethical statement

The present observational study was a part of the COVID-19 management directives as per the Government of India [3]. The ethical permission for the study has been obtained from the institute ethical committee (Letter no. Dean/2021/EC/2659).

### Existing IPC program

Before the COVID-19 pandemic, no well-constituted hospital infection control committee (HICC) was in existence except for a designated Infection Control Officer (ICO). The last institutional meeting on IPC had taken place in 2018, November. However, under the guidance of the ICO, a few on-training clinical microbiologists from the Department of Microbiology had set an Infection Control Team (ICT). The team continued its effort towards containment of infections through various activities which included outbreak surveillance, sensitization on prioritization of surface disinfection in hospitals vis-à-vis diminishing significance of routine microbiological surveillance of hospital environment, and passive antimicrobial resistance surveillance including antimicrobial stewardship. No regular infection control educational programs in form of continuous medical education (CME) or meetings were conducted since the beginning of 2019. In June 2019, initiatives for fresh sessions on infection control in form of lectures and demonstrations on hand hygiene, biomedical waste disposal,

**Table 1. Timeline of the IPC training activities in the study centre at the beginning of the COVID-19 pandemic.**

| Date | Topics of CME | Target population | Status of COVID-19 | References |
|---|---|---|---|---|
| 3rd December2019 | HH, NSI, BMWM | D | None | |
| 5th December2019 | AMSP | D | None | |
| 19th December 2019 | HH, BMWM | HS | Cluster of cases of pneumonia of unknown origin occurring in China | Zhou et al [4] |
| 31st December 2019 | HH, BMWM | N | WHO China office was informed of cases of pneumonia of unknown etiology detected in Wuhan City | WHO [5] |
| 7th January 2020 | HH, BMWM | N | | |
| 8th January 2020 | HH, BMWM | N | China identified the unknown pathogen as a new type of coronavirus. | WHO [6] |
| 21st January 2020 | EC | D, N | First evidence of human to human transmission | WHO [5] |
| 31st January 2020 | BMWM | HS | First positive case reported in India who travelled from Wuhan. | Andrews et al [7] |
| 12th February 2020 | HH, BMWM | D, HS | | |
| 14th February 2020 | SP | D, HS | | |
| 25th February 2020 | HH, BMWM | N | | |
| 29th February 2020 | HH, BMWM | N | | |
| 4th March 2020 | COVID-19 specific training | D, N, HS | Setting up of Task force for IPC in COVID-19 | |

HH = Hand Hygiene, NSI = Needle stick injury, BMWM = Bio-medical Waste Management, AMSP = Anti-microbial Stewardship Program, EC = Environmental Cleaning, SP = Standard Precautions, D = Doctors, N = Nurses, HS = Housekeeping Staff.

surface disinfection, needle stick injuries prevention and management, and introduction to antimicrobial stewardship program were initiated to update the healthcare personnel (HCP) based on the recently established national and international guidelines by the ICT. The interactive classes and infection control meetings were gradually regularized much ahead of the beginning of the recent outbreak. The timeline of these events has been shown in Table 1.

### Inclusion of the ICT in the COVID-19 task force

As per the government directive [3], a task force was to be set up for the imminent pandemic. The task force would be responsible for the creation of a COVID-19 facility within the existing infrastructure. Implementation of IPC practices were one of the basic requirements in this facility which necessitated the inclusion of the ICO. Following this, an urgent meeting was called by the medical superintendent of the hospital involving heads of the various departments of medical specialties, intensivists, laboratory personnel, nursing superintendents, ICO, and others. The ICO recommended expanding the ICT for providing best practices in the form of standard operating procedures (SOPs) and daily activity checklists, raising awareness amongst the personnel deployed in the facility. A core team was constituted by the ICO with the existing ICT members, designated infection control nurses, and post-graduate trainees of the Department of Community Medicine and Department of Pharmacology.

### The implementation of IPC in the COVID-19 hospital

The ICT planned and executed a phase-wise approach which comprised of the following steps (Fig 1)

**Assess.** The team would assess the existing level of maturity of IPC practices and identify the gaps. A general assessment of the knowledge, attitude, and practices was made from the pre- and post-class questionnaires from the previously conducted classes on infection control.

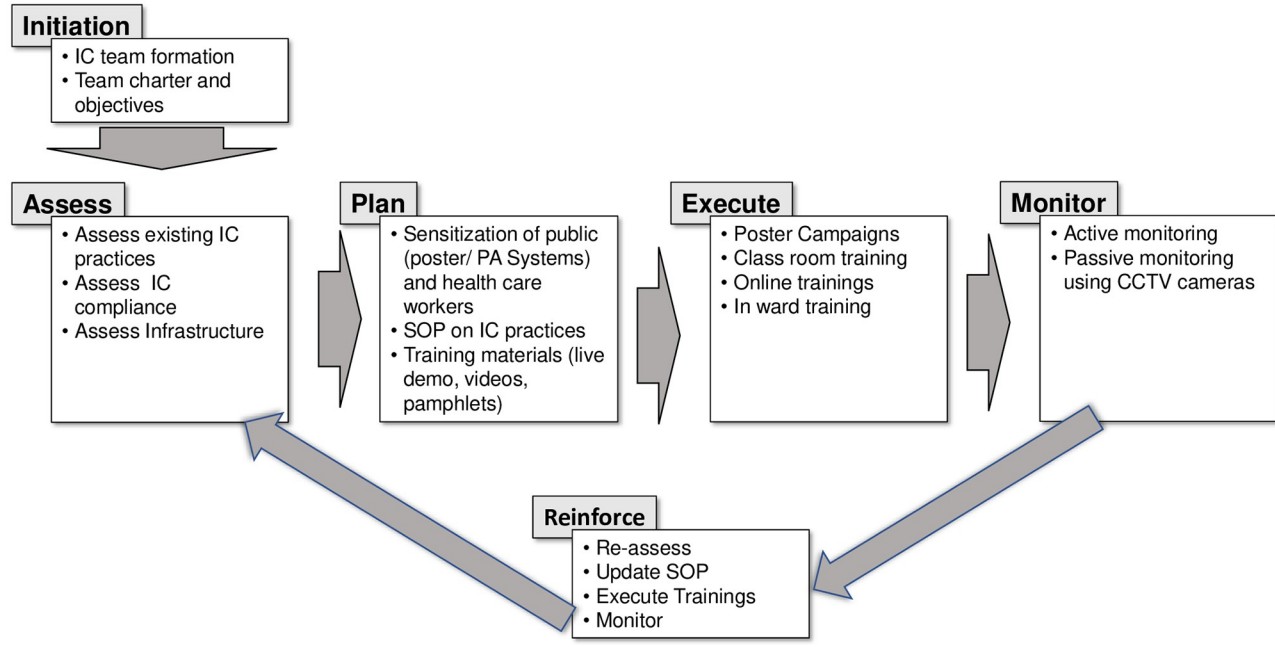

**Fig 1. Approach to IPC implementation with specific activities at different phases of implementation.**

The infrastructure of the hospital setup was analyzed in terms of layout for proposing designated areas and zones for various IPC to ensure the minimum spread of infection, types of doors like swinging versus lock mechanism, availability of water supply, air conditioning, and available ventilation. The ICT conducted multiple rounds of the entire hospital and interviewed the HCP in particular areas for their suggestions to make the physical environment conducive for implementing IPC practices based on the available resources and the type of planned interventions.

**Plan.** The priority of the team was to sensitize the HCP including doctors, nurses, paramedics, housekeeping and sanitation staff as well as the public in and around the hospital in form of poster campaigning, classroom training, online training, on-site (COVID wards) training. A crucial step in planning for the training was to identify the roles of a demonstrator, instructor, and moderator along with the target audience for each of the training sessions. The team designed training materials in form of posters, public announcement systems, demonstration videos, and SOPs on the IPC policies adopted for COVID-19 in the hospital following national and international guidelines. Areas in the hospital with maximum footfall were identified for putting electronic displays and posters to raise awareness about the ongoing pandemic. The next step in the planning was to demarcate the hospital's physical environment which was not structured for easy implementation of IPC practices. The existing facility was to be demarcated as the patient area (red zone) for housing COVID-19 patients and the clean area (green zone) for HCP with adequate protective barriers. Further, issues like the movement of staff and patients; biomedical waste (BMW) handling needed special attention.

**Execute.** The first sensitization on COVID-19 was conducted on 5th March 2020 (exactly 14 days before the first COVID-19 case was reported from Varanasi). This session emphasizing the importance of IPC practices was planned for the nursing in-charges of various sections of the hospitals and sanitary supervisors for spreading the awareness. The content of the training was primarily derived from the existing infection control training materials on hand hygiene,

social distancing, respiratory hygiene, surface disinfection, use of personal protective equipment, waste and linen management, transmission-based precautions including appropriate use of masks, and dead body management [8, 9]. For the initial 3–4 weeks, only physical training sessions were conducted. Following the announcement of the nationwide lockdown from March 25th 2020, the focus was also shifted to conducting online sessions using the telemedicine facilities available in the hospital. However, in-person training with limited participants maintaining physical distancing was continued in most of the instances. Video snippets prepared by the ICT were disseminated using social networking platforms. From the beginning, repeated training was targeted for better compliance. After the initial few sessions, the subsequent sessions consisted of fresh and trained participants. Native language was chosen as the medium of communication in the interactive training sessions.

A 300 bedded newly constructed and yet to be used super-specialty block was made the designated COVID-19 level 3 facility for those COVID-19 patients referred from primary and secondary COVID-19 hospitals for advanced care. The entire area inside the hospital building was divided into two color-coded zones: the red zone comprising of the intensive care units (ICU) or wards for the COVID-19 patients, hand washing, doffing areas whereas the green zone comprised of the nursing stations, the closed-circuit television (CCTV) monitoring rooms, resting rooms and donning areas for the HCP. Hand washing areas were not available at several wards which were constructed on instruction along with placement of hand hygiene stations within the ICU and wards. The floor-wise demarcations of the hospital building areas into green and red zones after planned modifications have been depicted in Fig 2. Arrow diagrams were also placed on the walls to guide the patient and staff traffic inside the building. The electronic and physical posters (displaying the Dos and Don'ts in the context of COVID-19 based on the available information at the time) were placed in the previously identified areas. To promote physical distancing, the waiting areas and other areas of the hospital were redesigned which included blocking alternate chairs in sitting areas, placing physical barriers, blocking unnecessary passages to restrict movement.

**Monitor.** Following training, monitoring of the HCP for the practice of IPC during their duties were done by direct observation and CCTV monitors installed in the duty areas. Onsite monitoring of the checklists for cleaning, disinfection was also performed by the ICT, and data on biomedical waste disposal, linen management were also recorded in registers.

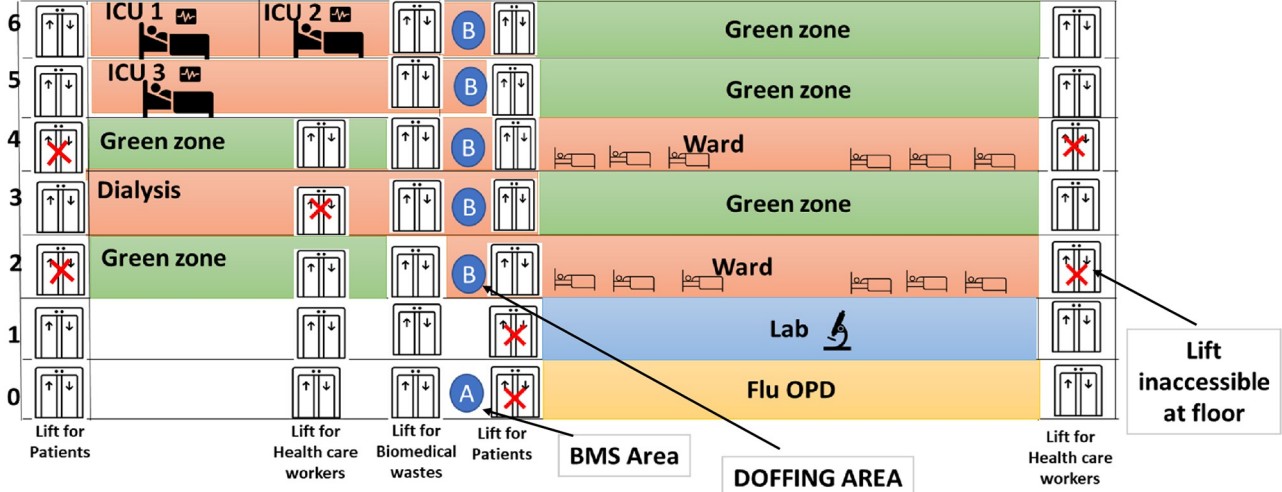

**Fig 2. Floor wise plan of the dedicated COVID-19 treatment centre into different zones for ease of IPC implementation.**

**Reinforce.** To cater to the evolving situation and the continuous updates from various national and international agencies, the team decided to have these steps repeated in cycles. The adopted policies in the SOPs were modified in subsequent revised versions following the changes in the recommendation from national and international guidelines. The frequency of execution of the loop and the duration would be initially rapid and then gradually regularized as time progresses.

## Audits in IPC

Regular auditing of the cleaning practices was carried out through visual assessment during scheduled and surprise onsite visits in the patient care areas by the ICT and assessment of the checklists and records. Environmental cleaning audit score sheets were used to document the overall compliance. Compliance was measured as the percentage of surfaces cleaned against those expected to be cleaned.

Record keeping for BMW was introduced and monitored. IPC training at least once before duty in the COVID-19 facility and preferably repeat training were prioritized and recorded. Hand hygiene compliance by monitoring the 5 moments of hand hygiene based on visual observation of HCPs was done by visits of the ICT as well as CCTV monitoring. A qualitative assessment of the viewpoints of the HCP was done based on the interviews and the feedback questionnaire filled by the participants at the end of the training sessions.

## Results

### HCP training

From March 2020 to June 2021, a total of 2552 HCP were trained and re-trained through 104 offline/onsite training sessions. Additionally, 548 medical students and interns were trained for IPC in COVID-19 through 23 online sessions which also involved demonstration through videos prepared by the ICT. Fifty-two training sessions were conducted based on 'buddy system' training for the on-duty group of HCP in presence of the ICT. Personal skills of the individual members were also explored to assign them with specific tasks like preparation of online video materials, data compilation and audit, onsite demonstration and monitoring, etc. based on their expertise. The initial weekly developments in the implementation of IPC practices in the hospital along with the rising total number of cases in India were depicted in Fig 3 along with the cumulative number of HCP who received training. With the growing evidence, the SOPs on the IPC policies in the hospital needed to be revised four times within a span of just 6 weeks. By the time the total number of COVID-19 cases reached the one thousand mark, close to a thousand HCP were trained on the IPC practices against COVID-19. At every session, the ICT attended to all the queries of the HCP related to IPC and beyond.

### Compliance with IPC practices

The overall compliance to the adopted IPC practices as evident from the audits was found to vary. Fig 4 shows the fluctuations in the compliance to surface disinfection over time against reinforcement training. It was noteworthy that at the beginning of the audit the overall compliance was 50% which increased to 80% and above. Despite the regular training, compliance decreased whenever newly recruited HCP were posted. With subsequent reinforcement training, there was a steady increase in overall compliance over the successive weeks. A rapid improvement in compliance was observed when the transmission dynamics of the SARS--CoV-2 were introduced as a part of the training content since the third week of May 2020. On comparing the compliance at two different periods in line with the emergence of the first and

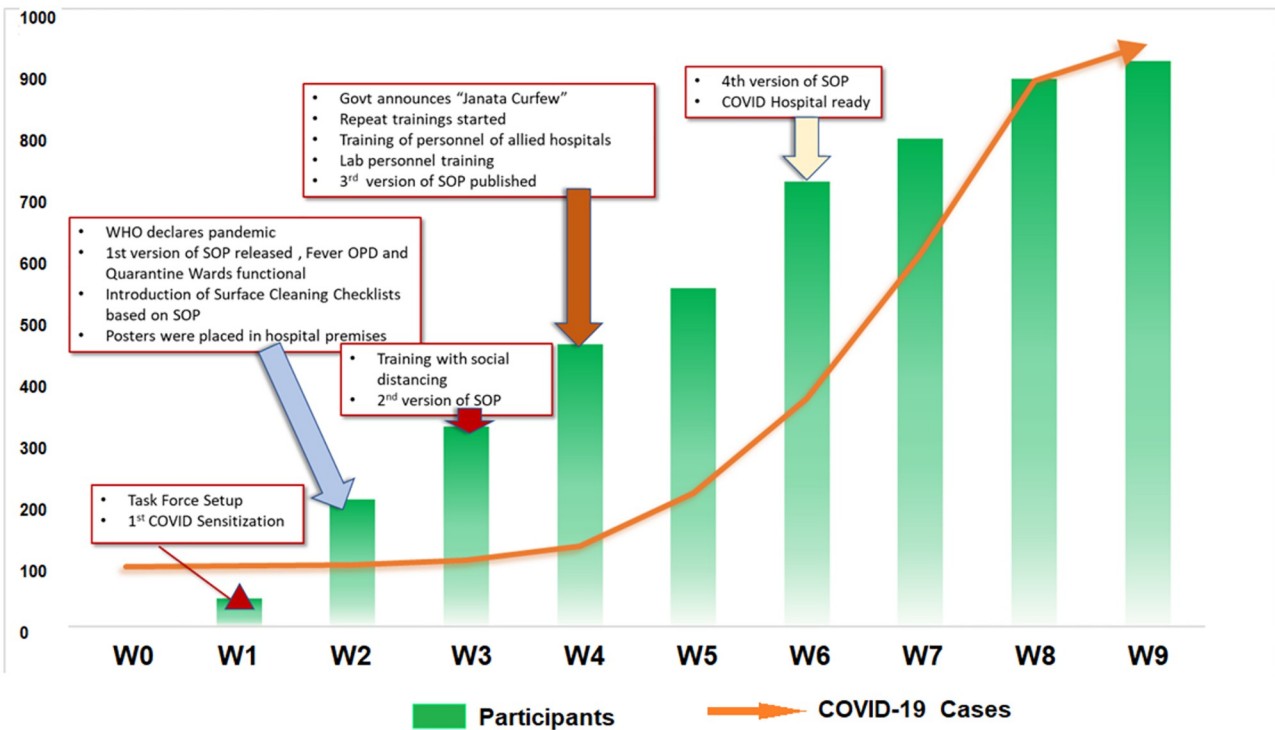

**Fig 3. Weekly advancement in the IPC training activities at the study centre and growing number of COVID-19 cases in the country at the beginning of the pandemic (W0 in the figure stands for the first week of the month of March 2020).**

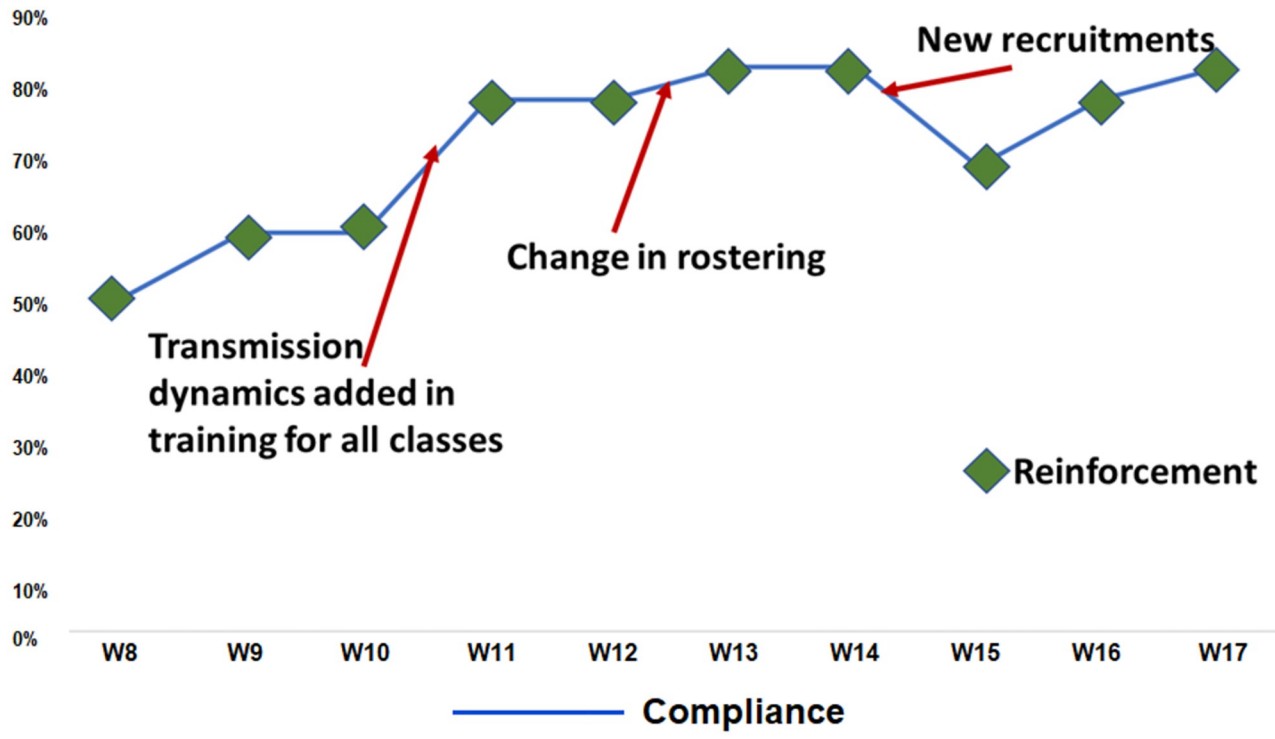

**Fig 4. Compliance trends to the surface disinfection related activities (W8 in the figure stands for the first week of the month of May 2020).**

second wave in India, no significant difference was seen. BMW records in form of the number of specific color-coded bags generated daily, movement of these wastes, and time of final disposal were fully maintained (100%). However, compliance to segregation of waste was only 50%. Hand hygiene compliance was poor varying from 10 to 40% complete adherence rate. During the training sessions, we could identify few staff members among their colleagues with leadership qualities who could motivate their fellow colleagues to adhere to the hand hygiene moments during duty hours. However, the major limitations for hand hygiene were false perception of protection with gloved hands and failure to perform hand hygiene following exposure to the patient's environment. None of the HCP was posted for duty without training on IPC and without confirmation by the ICT. The zoning of the existing hospital into green clean areas and red contaminated areas were easily understood by different strata of healthcare givers as evident by no major breach in the flow of movement during the study period.

### Challenges and facilitators in IPC implementation

The visits provided the opportunity for the interaction of the ICT members with the clinical staff and others at their workplaces to address their views and concerns. As revealed from the interviews and the surveys, the fear psychosis in the pandemic seemed to be the greatest facilitator for adopting the IPC practices. However, there were different internal and external factors posing challenges to the implementation of the IPC. Among the internal factors, inefficient duty rostering of the untrained housekeeping and sanitary staff often led to dissatisfaction among the HCPs. Most of the housekeeping staff was employed temporarily without any service benefits from the employer. Wearing PPE for long duty hours and continuous posting in the areas with high transmission risk led to a lack of motivation in the HCP which together with already existing low compliance to the IPC practice before the pandemic became a major hurdle for the IPC implementation. Different conflicting guidelines from national and international bodies and misinformation floating over the social media platforms were the other impediments to the task. However, the COVID-19 hospital and the IPC practices initiated were sustained throughout the three major waves due to variants of the virus in India (till January 2022).

The entire work has been summarized in the Fig 5.

## Discussion

With millions of infections, the COVID-19 pandemic has affected almost every state of wellbeing across the globe. While out of proportions losses of life and livelihood were seen on one hand, the pandemic also provided opportunities for implementation of basic healthcare infrastructure and practices, especially in resource-limited settings. In this regard, we describe the implementation and sustenance of an IPC program that was initiated in the pandemic in a major tertiary care hospital in north India.

An IPC program with an IPC team of dedicated members should exist in a hospital to mount an effective and timely response to the COVID-19 pandemic following the recommended strategies and practices [10]. Most of the hospitals across India lack a robust infrastructure for IPC for decades [11]. The deficiency became entirely apparent with the emergence of the COVID-19 pandemic and required immediate action. A recent worldwide survey on IPC practices had revealed that a dearth of trained staff, infrastructure, and resources as the major barriers [12]. Fortunately, in our hospital the administrative authorities recognized the importance of scaling up the IPC activities and provided the required logistic and managerial support on a priority basis even before the admission of the first laboratory-confirmed COVID-19 case in our hospital. The inclusion of the ICT in the COVID-19 task

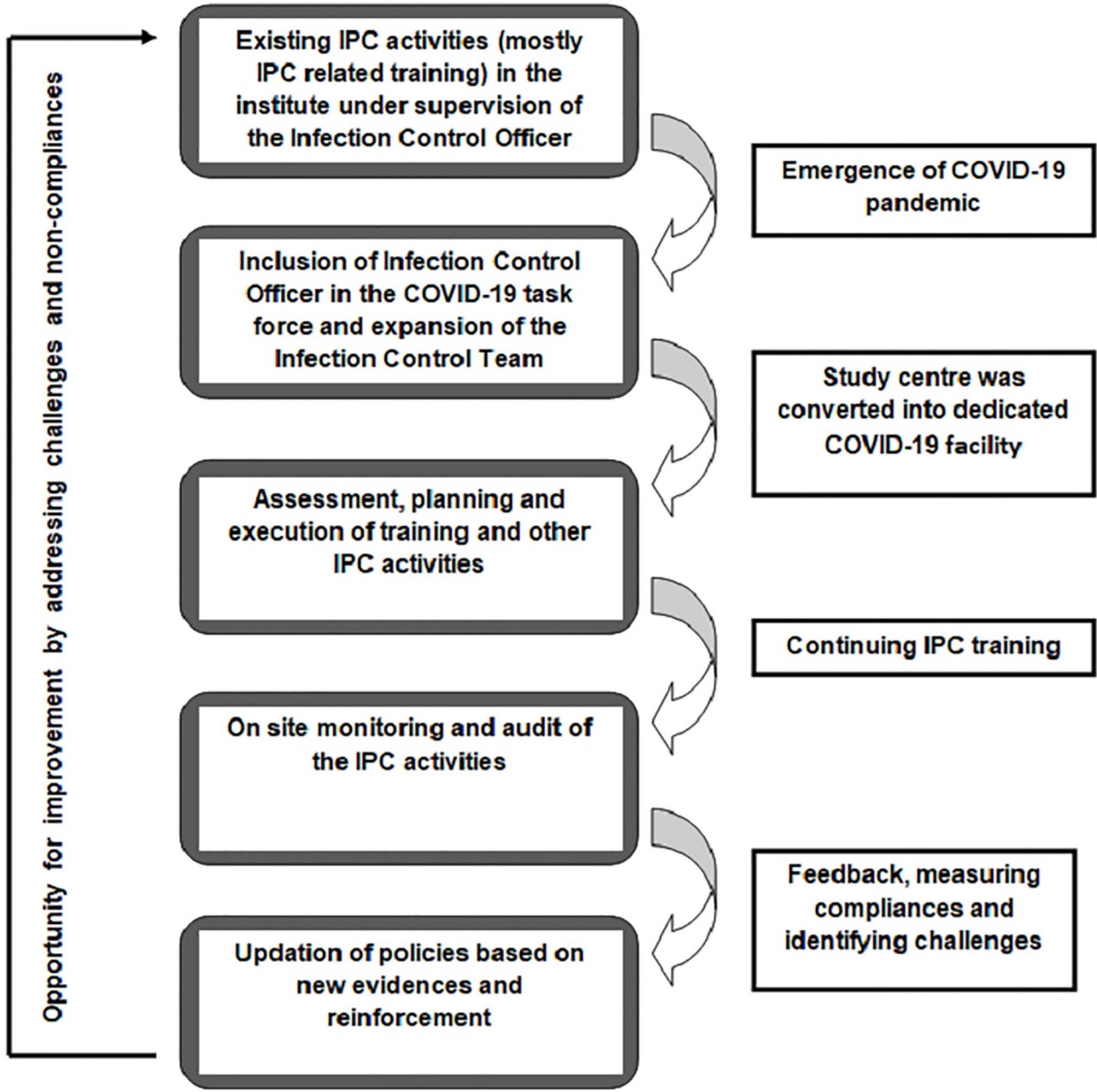

**Fig 5. Summary of work conducted in the study.**

force of the institute was the first welcoming step in the context. Published literature from one of the premier hospitals in the Kingdom of Saudi Arabia supports this action with the inclusion of multiple teams in the COVID-19 task force under the Infection Control and Prevention Department [13]. The same report also mentioned that volunteering opportunities were announced to double the IPC manpower. We also experienced that the planned activities under the IPC were beyond the capacity of the existing team and the team was expanded to include volunteering PG trainees from various departments. The recruits were first trained by expert ICT members under the guidance of the ICO. One of the important but frequently

overlooked reasons for non-adherence to the IPC practices is suboptimal role models [14]. The influence of role models for IPC in the context of hand hygiene is well documented in the literature [15, 16]. During our training sessions, we could identify the popular and influential staff among their colleagues and encourage their leadership during duty hours.

The architectural design of a hospital and its effect on patient and staff safety received considerable attention recently, but mostly in developed countries [17]. The very fact that the majority of the Indian hospitals are not designed for the management of an airborne infection SARS-CoV-2 per se, cannot be turned down in the first place. Construction and designing a new structure or dismantling and renovation of the existing buildings as per the IPC requirement especially at the time of lock-down seemed to be a mammothian task for a country of 14 billion people with 0.5 hospital beds per thousand populations [18]. Instead, we focussed on improvising the already existing infrastructure as a part of the rapid response of equipping the existing hospital with the requirement of managing COVID-19 patients. The zoning into green clean areas and red contaminated areas were meticulously done and implemented with feedback from different strata of healthcare givers enabling them adequate comfort in the workplace during the stressful duty hours at the same time not compromising the stringent IPC policies adopted for the hospital.

During the COVID-19 pandemic, hospitals across the globe struggled to maintain adequate staffing. This was also true for the most developed nations [19]. While the high workload and understaffing had direct negative implications for infection control practices, training of the recruits was critical to the successful implementation of IPC norms and also time taking [11]. In our experience, we observed that the compliance decreased when newly recruited HCP was posted to manage the increased workloads. Nevertheless, the decline in compliance seemed to be a temporary event and the compliance improved with ongoing training sessions.

The frontline healthcare workers were vulnerable to mental health disorders while working in a stressful work environment with continuous fear of acquiring the fatal infection [20]. To address this issue, the WHO and other health organizations have advocated for rotation of the HCP from higher stress areas to lower and vice versa [21]. Ministry of Health and Family Welfare (MoHFW), Government of India in collaboration with National Institute of Mental Health and Neurosciences (NIMHANS), Bengaluru, India had also published guidelines recommending the same [22]. We also observed that a serious lack of motivation to follow the IPC practices whenever HCP were posted for consecutive days in areas with the high-risk transmission of SARS-CoV-2. Additionally, the key stakeholders in an effective IPC program are the housekeeping staffs [11], who in our case was mostly recruited temporarily on an urgent basis and were untrained. Although, the high rate of HCP turnover was considered to be a major barrier in the IPC implementation [11], the paradoxical consequence of limiting the staff replacement during the stressful COVID-19 situation must be judged with caution.

While conducting the cleaning audit, the ICT followed the visual assessment method. Although the visual assessment based on the appearance of an item or surface against a checklist of standards is the most frequently used method for auditing disinfection and cleaning in hospitals, it is considered an inferior indicator of cleaning and disinfection [23]. However, the visual assessment method seemed to be cost-effective especially in resource-constrained settings [24]. Moreover, we considered the visual assessment method to be fast enough for rapid assessment of all surfaces in the ward requiring immediate attention. Our argument could also be supported with some published literature that described the method in question as more appropriate from a quality control perspective [25].

One of the important realizations for the IPC leadership in our hospital was that inclusion of transmission dynamics of the SARS-CoV-2 in the training sessions universally for all strata of HCP rapidly improved the HCP compliance. Understanding the mode of infection seemed

to be convincing to the HCP, the importance of every action prescribed in the SOPs. Continuous dialogue is of utmost importance as HCPs have a lot of expectations and dependence on the ICT [26]. Virtual meeting websites became the mode for teaching, discussion, and troubleshooting at times of restricted movement and gatherings. The online video materials served as attractive learning tools and were highly appreciated by different strata of HCP including housekeeping staff, security personnel, ambulance drivers, etc. The videos available in the ICT's own YouTube channel not only guided the HCP as ready references at workplaces but also disseminated the evidence-based knowledge on basic prevention practices to tackle COVID-19 in the community. But the challenge remained to filter out the misinformation floating over the social media which easily influences the HCP and general public at large to adopt practices against the medical evidence.

The study was not without limitations. We could not monitor or audit every aspect of IPC practices. For example, specific quantitative data on basic practices like hand hygiene compliance was lacking. This was mostly due to a negligible number of infection control nurses (only 1 for 1500 beds) in the hospital. Nonetheless, the opportunity of implementation of IPC for COVID-19 was not missed and a well-constituted, planned, and active task force came into existence that is continuing its activities. The hospital realized the importance of IPC in daily practice.

Although with worldwide effort, different vaccines have come in existence as a potential measure of prevention of COVID-19 disease, their variable potency and short-lived immunity have resulted in vaccine breakthrough infections [27]. The evolving variants of concern further has complicated the scenario causing subsequent devastative waves of the pandemic taking many precious lives all over the world. Under such circumstances, (IPC)practices in a hospital remain imperative for all individuals including the vaccinated healthcare personnel and the patients [28]. In this regard, our endeavour with continuing improvements is expected to be instrumental throughout the future uncertain period of COVID-19 pandemic.

## Conclusion

The study revealed the methods followed for implementation of IPC in an upcoming COVID-19 hospital in India and the major challenges faced in this regard. The findings of this study can serve as a basic platform for resource limited setups with similar challenges to implement IPC. The study clearly recognised the role of an infection control team, repeated training, monitoring and improvisation of the existing resources which were the keys for successful implementation of IPC practices in hospitals during the COVID-19 pandemic.

## Acknowledgments

The authors would like to thank the Medical Superintendent and Nodal officer, COVID-19 Taskforce, BHU for their support. We would also like to thank Dr S K Mathur, Dr S K Gupta, Dr Shampa Anupurba, Dr Jaya Chakravorty, Dr Sangeeta Kansal, Dr Rajeev Dubey, Dr Amit Singh, and residents of the Department of Microbiology for their assistance.

## Author Contributions

**Conceptualization:** Tuhina Banerjee.

**Data curation:** Arghya Das, Rahul Garg, Dharanidhar Singh, Bisweswar Ojha, H. Larikyrpang Kharchandy, Bhairav Kumar Pathak, Pushkar Srikrishnan, Ravindra Singh, Immanuel Joshua, Sanket Nandekar, Vinothini J., Reenu Reghu, Nikitha Pedapanga, Kamal Kumar Yadav.

**Formal analysis:** Arghya Das, Rahul Garg, Tuhina Banerjee.

**Investigation:** Arghya Das, Rahul Garg, E. Sampath Kumar, Dharanidhar Singh, Bisweswar Ojha, H. Larikyrpang Kharchandy, Bhairav Kumar Pathak, Pushkar Srikrishnan, Ravindra Singh, Immanuel Joshua, Sanket Nandekar, Vinothini J., Reenu Reghu, Nikitha Pedapanga, Kamal Kumar Yadav.

**Methodology:** Arghya Das, Tuhina Banerjee.

**Project administration:** Tuhina Banerjee.

**Resources:** Tuhina Banerjee.

**Software:** E. Sampath Kumar, Dharanidhar Singh.

**Supervision:** Tuhina Banerjee.

**Validation:** Tuhina Banerjee.

**Writing – original draft:** Arghya Das, Rahul Garg.

**Writing – review & editing:** Tuhina Banerjee.

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
