## [Decision Letter · Decision Letter 0]

27 Jan 2022

PONE-D-22-00130Implementation of infection prevention and control practices in an upcoming COVID-19 hospital in India: An opportunity not missed.PLOS ONE

Dear Dr. Banerjee,

Thank you for submitting your manuscript to PLOS ONE. After careful consideration, we feel that it has merit but does not fully meet PLOS ONE’s publication criteria as it currently stands. Therefore, we invite you to submit a revised version of the manuscript that addresses the points raised during the review process.

We look forward to receiving your revised manuscript.

Kind regards,

Sanjay Kumar Singh Patel, Ph.D.

Academic Editor

PLOS ONE

Journal Requirements:

Reviewers' comments:

Reviewer's Responses to Questions

**Comments to the Author**

1. Is the manuscript technically sound, and do the data support the conclusions?

Reviewer #1: Yes

Reviewer #2: Partly

Reviewer #3: Yes

2. Has the statistical analysis been performed appropriately and rigorously? 

Reviewer #1: N/A

Reviewer #2: N/A

Reviewer #3: Yes

3. Have the authors made all data underlying the findings in their manuscript fully available?

Reviewer #1: Yes

Reviewer #2: Yes

Reviewer #3: Yes

4. Is the manuscript presented in an intelligible fashion and written in standard English?

Reviewer #1: Yes

Reviewer #2: Yes

Reviewer #3: Yes

5. Review Comments to the Author

Reviewer #1: In this paper entitled "Implementation of infection prevention and control practices in an upcoming COVID-19 hospital in India: An opportunity not missed", the authors investigated the deficiencies in existing IPC practices with the help of review of checklist and visual inspection. 2552 HCP and 548 medical students were trained in IPC through multiple offline/onsite sessions. The results showed that the overall compliance to surface disinfection and cleaning increased from 50% to >80% with repeated training. However, the compliance decreased whenever new recruits were posted. The manuscript recognizes the role of successfully implementing IPC practices in the hospital during the COVID-19 pandemic. The manuscript is easy to understand and well written. Although, the manuscript has no technical basis for rejection and can be accepted for publication. But, it has few minor problems.

Minor Comments:

1) The only major issue with the manuscript is the presentation of the manuscript. There is no issue with the English of the manuscript. But the manuscript could be better organized. For example, it isn't very easy for the reader to differentiate between the content of Materials and Method section and Results section.

2) Introduction: Minor information on the variants of COVID-19 and their future challenges can be included i.e. doi: 10.1007/s15010-021-01734-2.

3) It would be required to provide one illustrative figure to summarize the whole study.

4) The authors should cross-check all abbreviations in the manuscript. Initially, define in full name followed by abbreviation.

Reviewer #2: This manuscript discusses about Implementation of infection prevention and control practices in an upcoming COVID-19 hospital in India. The manuscript needs following major changes to improve it further.

Comments:

•The manuscript is fairly excellent, but greater effort should be made to provide and debate current published studies on covid-19 waves and variations, as well as infection prevention and control techniques.

• The last institutional meeting on IPC had taken place 81 in 2018, November. However, under the guidance of the ICO, a few on-training clinical 82 microbiologists from the Department of Microbiology had set an Infection Control 83 Team (ICT). [If any reference available, please cite it]

• In June 2019, initiatives for fresh sessions on 90 infection control in form of lectures and demonstrations on hand hygiene, biomedical 91 waste disposal, surface disinfection, needle stick injuries prevention and management, 92 and introduction to antimicrobial stewardship program were initiated to update the 93 healthcare personnel (HCP) based on the recently established national and international 94 guidelines by the ICT. [If any reference available, please cite it]

• In table number 1 add references as another column.

• In the conclusion, add a few additional lines to better address the topic properly. In short, how your work is more valuable and novel than other reported studies till date.

• Improve quality of figures.

Reviewer #3: The manuscript is well written, and it can be accepted after the minor revision. Please find my comments below.

1. Please provide some detailed information about COVID-19, it’s prevention strategies including role of health, immunity, and natural biomolecules i.e., doi.org/10.1007/s12088-020-00908-0;
doi.org/10.1007/s12088-020-00893-4

2. Introduction, please include some quantitative information about cases, mortality, and casualty of COVID-19.

3. Discussion, please highlight minor information about COVID-19 variants and their challenges in its prevention.

---

## [Author Response · Author response to Decision Letter 0]

9 Mar 2022

Comments by the Reviewer 1

1) The only major issue with the manuscript is the presentation of the manuscript. There is no issue with the English of the manuscript. But the manuscript could be better organized. For example, it isn't very easy for the reader to differentiate between the content of Materials and Method section and Results section. 

Response: Complied.

The authors are grateful to the reviewer for his insightful remark which has helped us to organize the significant portions of the manuscript in order to make it easily understandable by the readers.

Changes are highlighted in lines 173-189 of the revised manuscript.

2) Introduction: Minor information on the variants of COVID-19 and their future challenges can be included i.e. doi: 10.1007/s15010-021-01734-2. 

Response: Complied. 

The authors want to thank the reviewer for the suggestion. The same issue has been raised by the other reviewers and addressed briefly under the discussion section.

Additions are highlighted in lines 383-391 of the revised manuscript.

3) It would be required to provide one illustrative figure to summarize the whole study. 

Response: Complied.

A new illustrative Figure 5 in form of a flow diagram has been added and also cited in the revised manuscript (Lines 276-278).

4) The authors should cross-check all abbreviations in the manuscript. Initially, define in full name followed by abbreviation. 

Response: Complied.

We have checked all abbreviations and as directed, all abbreviations have been expanded when they have mentioned first in the manuscript.

Comments by the Reviewer 2

•The manuscript is fairly excellent, but greater effort should be made to provide and debate current published studies on covid-19 waves and variations, as well as infection prevention and control techniques. 

Response: Complied.

Although the work of the manuscript was carried out during the period of the pandemic when variations in the virus was not known, we endeavored to mention the hardships which might be posed from infection control point of view due to the emerging variants in the subsequent waves under discussion section to comply with the reviewer’s suggestion.

• The last institutional meeting on IPC had taken place in 2018, November. However, under the guidance of the ICO, a few on-training clinical microbiologists from the Department of Microbiology had set an Infection Control Team (ICT). [If any reference available, please cite it] 

Response: As such, there is no reference. 

This was an intra-institutional meeting. The copy of the circular of the mentioned meeting may be produced on case to case basis on request.

• In June 2019, initiatives for fresh sessions on infection control in form of lectures and demonstrations on hand hygiene, biomedical waste disposal, surface disinfection, needle stick injuries prevention and management, and introduction to antimicrobial stewardship program were initiated to update the healthcare personnel (HCP) based on the recently established national and international guidelines by the ICT. [If any reference available, please cite it] 

Response: The mentioned activity was conducted under the intra institutional teaching activities by the Infection Control Team.

The copy of the teaching schedule/ attendance records may be produced, if required.

• In table number 1 add references as another column. 

Response: Complied. 

The authors want to thank the reviewer for this important comment.

Accordingly in Table 1, a separate column has been added with references of different COVID-19 timeline events coinciding with the IPC training activities in the institute.

The additions are highlighted within the table.

• In the conclusion, add a few additional lines to better address the topic properly. In short, how your work is more valuable and novel than other reported studies till date. Response: Complied.

Following the important suggestion of the reviewer, we have re-written the conclusion.

Changes are highlighted in lines 394-400 of the revised manuscript

• Improve quality of figures. 

Response: Complied.

The quality of the figures has been substantially improved as per PLoS ONE requirements.

Comments by the Reviewer 3

1. Please provide some detailed information about COVID-19, it’s prevention strategies including role of health, immunity, and natural biomolecules i.e., doi.org/10.1007/s12088-020-00908-0;
doi.org/10.1007/s12088-020-00893-4

Response: The authors would like to appreciate the suggestion by the reviewer.

However, the authors feel that the present manuscript specifically focuses on the hospital infection prevention and control efforts made towards prevention and containment of COVID-19 disease in a dedicated COVID-19 treatment centre.

Although other strategies like immunization, biomolecules have substantial role in prevention, the same may not be relevant in context of the present manuscript.

2. Introduction, please include some quantitative information about cases, mortality, and casualty of COVID-19. 

Response: Complied.

The authors want to thank the reviewer for this valuable comment for enriching the manuscript.

Additions are highlighted in lines 57-60 of the revised manuscript.

3. Discussion, please highlight minor information about COVID-19 variants and their challenges in its prevention. 

Response: Complied.

The same has been addressed in response of the first comment of Reviewer 2.

Additions are highlighted in lines 383-391 of the revised manuscript.

---

## [Decision Letter · Decision Letter 1]

12 Apr 2022

PONE-D-22-00130R1Implementation of infection prevention and control practices in an upcoming COVID-19 hospital in India: An opportunity not missed.PLOS ONE

Dear Dr. Banerjee,

Thank you for submitting your manuscript to PLOS ONE. After careful consideration, we feel that it has merit but does not fully meet PLOS ONE’s publication criteria as it currently stands. Therefore, we invite you to submit a revised version of the manuscript that addresses the points raised during the review process. Please submit your revised manuscript by May 27 2022 11:59PM. If you will need more time than this to complete your revisions, please reply to this message or contact the journal office at plosone@plos.org. Please include the following items when submitting your revised manuscript:A rebuttal letter that responds to each point raised by the academic editor and reviewer(s). You should upload this letter as a separate file labeled 'Response to Reviewers'.A marked-up copy of your manuscript that highlights changes made to the original version. You should upload this as a separate file labeled 'Revised Manuscript with Track Changes'.An unmarked version of your revised paper without tracked changes. You should upload this as a separate file labeled 'Manuscript'.If applicable, we recommend that you deposit your laboratory protocols in protocols.io to enhance the reproducibility of your results. Protocols.io assigns your protocol its own identifier (DOI) so that it can be cited independently in the future. For instructions see: https://journals.plos.org/plosone/s/submission-guidelines#loc-laboratory-protocols. Additionally, PLOS ONE offers an option for publishing peer-reviewed Lab Protocol articles, which describe protocols hosted on protocols.io. Read more information on sharing protocols at https://plos.org/protocols?utm_medium=editorial-email&utm_source=authorletters&utm_campaign=protocols.

We look forward to receiving your revised manuscript.

Kind regards,

Sanjay Kumar Singh Patel, Ph.D.

Academic Editor

PLOS ONE

Journal Requirements:

Reviewers' comments:

Reviewer's Responses to Questions

**Comments to the Author**

1. If the authors have adequately addressed your comments raised in a previous round of review and you feel that this manuscript is now acceptable for publication, you may indicate that here to bypass the “Comments to the Author” section, enter your conflict of interest statement in the “Confidential to Editor” section, and submit your "Accept" recommendation.

Reviewer #1: (No Response)

Reviewer #2: All comments have been addressed

Reviewer #3: All comments have been addressed

2. Is the manuscript technically sound, and do the data support the conclusions?

Reviewer #1: Yes

Reviewer #2: Yes

Reviewer #3: Yes

3. Has the statistical analysis been performed appropriately and rigorously? 

Reviewer #1: Yes

Reviewer #2: Yes

Reviewer #3: N/A

4. Have the authors made all data underlying the findings in their manuscript fully available?

Reviewer #1: Yes

Reviewer #2: No

Reviewer #3: Yes

5. Is the manuscript presented in an intelligible fashion and written in standard English?

Reviewer #1: Yes

Reviewer #2: Yes

Reviewer #3: Yes

6. Review Comments to the Author

Reviewer #1: In this manuscript, “Implementation of infection prevention and control practices in an upcoming COVID-19 hospital in India: An opportunity not missed.” the authors investigated the infection prevention and control practices in COVID-19 hospital in Varanasi, India. The manuscript is relevant and vital for publication. The manuscript identifies deficiencies in the existing IPC practices and, via repeated IPC training helps to remove the shortcomings. Although the manuscript has exciting findings, but it has minor problems.

1) The manuscript is well written. The Abstract, Introduction, and Material & Method are nicely written. However, the results section is not extensively reported in the manuscript. There are few key points discussed in the Discussion section that can be moved to the Result section.

2) Please format the reference section according to Journal requirements.

Reviewer #2: Now, the authors have thoroughly revised the paper. Hence, the work is suitable for publishing and should be accepted.

Reviewer #3: Accept

7. PLOS authors have the option to publish the peer review history of their article (what does this mean?). If published, this will include your full peer review and any attached files.

Reviewer #1: **Yes: **Aditya Kumar Sharma

Reviewer #2: No

Reviewer #3: **Yes: **Dr. Deepak Kumar Padhi

---

## [Author Response · Author response to Decision Letter 1]

21 Apr 2022

1) The manuscript is well written. The Abstract, Introduction, and Material & Method are nicely written. However, the results section is not extensively reported in the manuscript. There are few key points discussed in the Discussion section that can be moved to the Result section.

Response to Comment: Complied.

We are grateful to the reviewer for his kind words of appreciation. His further instructions for the revised manuscript have also been complied. We have also shifted some of the points from Discussion to Results. 

Changes are highlighted in lines 254-257 and 260-263 of the revised manuscript.

2) Please format the reference section according to Journal requirements.

Response to Comment: Complied.

The authors want to thank the reviewer for the valuable suggestion.

The following changes have been made to the Reference section in the revised manuscript, as per the example given in the PLoS Instructions for authors.

i) The abbreviated names of the journals which were earlier italicized have been made non-italicized in the revised version.

ii) Month's name, which was mentioned after a year in few of the references, has been deleted from the references in the revised version.

iii) DOI i.e. Digital Object Identifier has been added in addition to traditional volume and page numbers of the cited references (as mentioned in the Submission Guidelines of PLOS ONE).

All changes are highlighted in yellow.

---

## [Editor Report · Decision Letter 2]

22 Apr 2022

Implementation of infection prevention and control practices in an upcoming COVID-19 hospital in India: An opportunity not missed.

PONE-D-22-00130R2

Dear Dr. Banerjee,

We’re pleased to inform you that your manuscript has been judged scientifically suitable for publication and will be formally accepted for publication once it meets all outstanding technical requirements.

Kind regards,

Sanjay Kumar Singh Patel, Ph.D.

Academic Editor

PLOS ONE

---

## [Editor Report · Acceptance letter]

12 May 2022

PONE-D-22-00130R2 

Implementation of infection prevention and control practices in an upcoming COVID-19 hospital in India: An opportunity not missed. 

Dear Dr. Banerjee:

I'm pleased to inform you that your manuscript has been deemed suitable for publication in PLOS ONE. Congratulations! Your manuscript is now with our production department. 

Kind regards, 

on behalf of

Dr. Sanjay Kumar Singh Patel 

%CORR_ED_EDITOR_ROLE%

PLOS ONE